# Plug-and-Display Photo-Switchable Systems on Plant Virus Nanoparticles

**DOI:** 10.3390/biotech11040049

**Published:** 2022-10-21

**Authors:** Louisa Kauth, Eva Miriam Buhl, Julian Luka, Karolin Richter, Ulrich Commandeur, Christina Dickmeis

**Affiliations:** 1Institute for Molecular Biotechnology, RWTH Aachen University, 52074 Aachen, Germany; 2Electron Microscopy Facility, Institute of Pathology, University Hospital RWTH Aachen, 52074 Aachen, Germany

**Keywords:** optogenetic systems, nanoparticles, photo-switches, plant viruses, LOVTRAP, BphP1/QPAS1, Dronpa145N

## Abstract

Light can be used to regulate protein interactions with a high degree of spatial and temporal precision. Photo-switchable systems therefore allow the development of controllable protein complexes that can influence various cellular and molecular processes. Here, we describe a plant virus-based nanoparticle shuttle for the distribution of proteins that can be released when exposed to light. Potato virus X (PVX) is often used as a presentation system for heterologous proteins and epitopes, and has ideal properties for biomedical applications such as good tissue penetration and the ability to form hydrogels that present signaling molecules and promote cell adhesion. In this study, we describe three different systems attached to the surface of PVX particles: LOVTRAP, BphP1/QPAS1 and Dronpa145N. We demonstrated the functionality of all three photo-switchable protein complexes in vitro and the successful loading and unloading of PVX particles. The new systems provide the basis for promising applications in the biomedical and biomaterial sciences.

## 1. Introduction

The behavior of many proteins is influenced by light, a phenomenon that can be exploited to create optogenetic systems in the biomedical and biomaterial sciences. For example, several protein pairs are known to interact reversibly in response to illumination at particular wavelengths and can thus form photosensitive switches [1]. These can be used to regulate and analyze biological processes, such as monitoring the shuttling of transcriptional factors [2], controlling the subcellular localization of proteins [3] or modulating protein activity in vivo [4]. Optogenetic systems can also be used to regulate gene expression [5] and to trigger protein crosslinking in hydrogels [6].

Several well-characterized optogenetic switch systems are currently available that respond to light in particular wavelength bands representing different parts of the visible and near-visible spectrum [7]. The light–oxygen–voltage trap and release of proteins (LOVTRAP) is an optogenetic switch based on protein dissociation triggered by blue light [8]. In the dark, the Zdark (Zdk) protein binds to LOV2, a photosensor domain of phototropin 1 from *Avena sativa* [9]. When the complex is exposed to blue light (400–500 nm), the Zdk protein is released (Figure 1A). LOV2 is a small, water-soluble protein that binds non-covalently to the flavin mononucleotide (FMN) cofactor [10]. When exposed to blue light, the LOV domain absorbs a photon, triggering a photoreaction in which a cysteine residue covalently binds FMN, resulting in a conformational change that causes the C-terminal Jα-helix to unfold, releasing Zdk [10,11]. In darkness, the LOV cysteine releases FMN and the Jα-helix folds back against the protein core, creating permissive binding conditions for Zdk [12].

Another system is based on the interaction between the phytochrome photoreceptor P1 (BphP1) from the bacterium *Rhodopseudomonas palustris* and protein QPAS1. BphP1 binds the cofactor biliverdin IXa, which absorbs near-infrared (NIR) light [13,14]. BphP1 can switch between two active forms: the Pr form absorbs red (600–700 nm) light, whereas the Pfr form absorbs far-red/NIR (740–780 nm) light [14]. A transition between these two forms occurs when biliverdin is irradiated and dimerizes, causing a conformational change in BphP1 that promotes or disrupts its interaction with other proteins [15]. The interaction between BphP1 and QPAS1 occurs exclusively when BphP1 is in the Pfr form, but is reversed when BphP1 is exposed to red light or darkness and switches to the Pr form (Figure 1B) [5]. The system has already been used to control gene expression and protein targeting [16].

Other systems involve self-assembly rather than interactions between protein partners. Dronpa is related to green fluorescent protein (GFP) but is ~2.5-fold brighter than enhanced GFP (eGFP). The monomeric protein consists of a β-barrel structure with an α-helix containing the chromophore. The mutant Dronpa145N exhibits light-dependent oligomerization and dissociation of the Dronpa domain. The protein is monomeric when excited with cyan light (500 nm), whereas irradiation with weak violet light (400 nm) causes the assembly of tetramers (Figure 1C). The combination of light-dependent and light-switchable behavior allows Dronpa145N to control the activity of fusion partners, for example, in the construction of photo-controlled hydrogels [6].

Optogenetic switches are ideal for the development of protein-based drug release systems when combined with a suitable carrier. In this context, plant viruses are nanoscale structures consisting of coat protein subunits that naturally self-assemble. The resulting plant virus nanoparticles (VNPs) are biocompatible, biodegradable and non-infectious in mammals, and allow the production of identical, symmetrical and monodisperse particles that can be synthesized in bulk using plants [17,18]. Plant VNPs have been modified to carry drugs, imaging reagents, immunomodulators, enzymes and biosensors [19,20,21,22,23,24,25,26]. Such applications are facilitated by the ability of some plant virus particles to penetrate tissues and accumulate naturally in tumors [27]. This property is influenced by the particle size, shape and distribution of surface charges, which affect the enhanced permeability and retention effect [28,29]. Indeed, VNPs based on filamentous plant viruses such as potato virus X (PVX) possess superior pharmacokinetic properties [30,31]. PVX is the type member of the genus *Potexvirus* in the family *Flexiviridae* [32,33]. It features a plus-stranded RNA genome of 6.4 kb [34] encoding five proteins [35]. The 515 × 15 nm helical PVX capsid has 1270 coat protein subunits assembled around the genomic RNA [35,36,37]. The N-terminus of the coat protein is presented on the outside of the assembled particle and can be genetically modified to incorporate additional amino acids and small peptides [37,38,39,40].

PVX has already been used as an in situ vaccine to induce antitumor immunity [41], as a drug delivery vehicle [42] and for the presentation of surface peptides [43]. Examples of the latter include the fluorescent proteins mCherry [44] and iLOV [45] as bioimaging tools. The combination of optogenetic switch proteins and plant VNPs offers an opportunity to develop protein-based therapeutics that can release drugs when illuminated with certain wavelengths of light. Here, we describe the use of the SpyTag/SpyCatcher (ST/SC) system [46] for the plug-and-display attachment of optogenetic switch proteins to the surface of modified PVX particles. One protein partner of the photo-switchable protein pairs LOVTRAP, BphP1/QPAS and Dronpa145N was fused to the SC domain, allowing its conjugation to the surface of PVX particles carrying the ST peptide. The second protein can then reversibly interact with its partner on the PVX surface when exposed to light, creating a novel photo-switching system for the attachment and release of molecules from VNPs. We found that PVX particle loading/unloading works successfully with the photo-switchable LOVTRAP, BphP1/QPAS1 and Dronpa145N systems, forming the basis for further applications in the fields of drug delivery, bioimaging and biomaterial design.

## 2. Materials and Methods

### 2.1. Expression and Purification of Proteins

Genes and cloning procedures are described in the Appendix A. The oligonucleotides used for cloning procedures are listed in Appendix A. The resulting fusion proteins were expressed in *Escherichia coli* BL21 star (DE3) cells (Thermo Fisher Scientific, Dreieich, Germany) by inoculating 5 mL lysogeny broth containing 100 µg/mL ampicillin with a single transformant colony and incubating, at 37 °C, overnight. A 1 mL aliquot of the pre-culture was used to inoculate 100 mL of the same medium, and the culture was incubated at 37 °C, shaking at 160 rpm, until the OD600 reached 0.5–0.8. Protein expression was then induced by adding isopropyl β-D-1-thiogalactopyranoside (IPTG) to a final concentration of 1 mM and incubating the cultures under the appropriate conditions for each construct (Table 1). The cells were then harvested by centrifugation (2500× *g*, 15 min, 4 °C) and the pellets were stored at –20 °C.

Bacterial pellets were resuspended in 5 mL of the appropriate lysis buffer (Table 2) for the target protein supplemented with 2 mg/mL lysozyme, and were incubated for 30 min on ice, shaking at 45 rpm. The cells were disrupted by sonication on ice (amplitude 60%, cycle 0.5) for 4 × 45 s with intervening pauses of 45 s. For lysis buffer not including any detergent, an additional incubation step was required in which we added 250 µL 20% (*v*/*v*) Triton X-100 and incubated the lysate for 30 min on ice, shaking at 45 rpm. The final lysate was centrifuged (16,000× *g*, 20 min, 4 °C), and the clear supernatant was retained.

Proteins with a His6 tag were purified by immobilized metal affinity chromatography (IMAC) using Ni-NTA agarose (Qiagen, Hilden, Germany), whereas proteins containing a glutathione S-transferase (GST) tag were purified using glutathione Sepharose 4 fast-flow resin (Cytiva, Marlborough, MA, USA). We equilibrated 1 mL of the resin with 3 × 4 mL lysis buffer appropriate for the target protein (Table 2) and once with 6 mL of the same lysis buffer before transferring the lysate to the column via a 0.45 µm syringe filter (Carl Roth, Karlsruhe, Germany) and setting aside the flow-through fraction. For the BphP1/QPAS and Dronpa145N proteins, the column was washed with 2 × 5 mL wash buffer appropriate for the target protein (Table 2). For LOVTRAP proteins, the column was washed with 2 × 5 mL low-salt and 3 × 5 mL high-salt buffers, and for LOV2 proteins, this was followed by an additional incubation step in 5 mg/mL FMN in low-salt buffer (Table 2). The proteins were eluted in a single step using 1 mL elution buffer containing imidazole or glutathione for the His6 and GST tags, respectively (Table 2). The eluates were concentrated and rebuffered using Vivaspin 6 columns with a molecular weight cutoff (MWCO) of 3–10 kDa (Sartorius-Stedim, Göttingen, Germany) in 50 mM Tris-HCl pH 8.5 containing 10% (*v*/*v*) glycerol.

### 2.2. ST/SC Reaction with Purified Components

We mixed 3 µg of purified PVX-ST particles with a two-fold molar excess of the SC-fusion protein in 50 mM Tris-HCl pH 8.5 and incubated the mixture overnight, at 4 °C. PVX-ST particles were purified as recommended by the International Potato Centre (Lima, Peru) with modifications ([48]. The samples were analyzed by sodium dodecylsulfate polyacrylamide gel electrophoresis (SDS-PAGE) using 4% stacking gels and 10% resolving gels. Each sample was mixed with 5 × loading buffer (62.5 mM Tris-HCl pH 6.8, 30% (*w*/*v*) glycerol, 4% (*w*/*v*) SDS, 10% (*w*/*v*) 2-mercaptoethanol, 0.05% (*w*/*v*) bromophenol blue) and heated to 100 °C for 5 min before separation at 180 V for 50–60 min in 1× SDS running buffer (25 mM Tris, 2 M glycine, 1% (*w*/*v*) SDS) alongside P7719 Color Prestained Protein Standards (New England Biolabs, Ipswich, MA, USA). The gels were stained with 0.25% (*v*/*v*) Coomassie Brilliant Blue G-250 in 50% (*v*/*v*) methanol, 10% (*v*/*v*) acetic acid.

For Western blotting, the proteins were transferred to a nitrocellulose membrane (Cytiva) in 1 × blotting buffer (25 mM Tris, 192 mM glycine, 20% (*v*/*v*) methanol) at 100 V for 1 h. The membrane was blocked with 4% (*w*/*v*) skimmed milk in phosphate-buffered saline (PBS) and incubated sequentially with the primary monoclonal antibody against His_6_ (Qiagen) or a polyclonal antibody against PVX from Deutsche Sammlung von Mikroorganismen und Zellkulturen (DSMZ, Braunschweig, Germany) or a polyclonal antibody against GST (produced in rabbits from the Institute for Molecular Biotechnology) and a secondary alkaline phosphatase (AP)-labeled antibody, each diluted 1:5000 in PBS. The membrane was washed with PBS (3 × 10 min) between each step. The signal was visualized with nitroblue tetrazolium chloride/5-bromo-4-chloro-3-indolylphosphate *p*-toluidine salt (Carl Roth) in AP buffer (100 mM Tris pH 9.6, 100 mM NaCl, 5 mM MgCl_2_). Coupling efficiency was estimated by the densitometric analysis of anti-PVX Western blots using ImageJ software. The standard deviation was determined from biological duplicates or triplicates.

### 2.3. Coupling ST/SC-Tagged Components in Crude Extracts and Purification of the Complexes

PVX-ST was produced in plants as previously described [48]. Plant material was harvested 18–21 days post infection (dpi), homogenized in two volumes (*w*/*v*) of 0.1 M phosphate buffer pH 7.0, filtered through three layers of gauze, and clarified by centrifugation (7800× *g*, 20 min, 4 °C). The SC-fusion proteins were expressed in *E. coli* BL21 star (DE3) cells, and lysates were prepared by sonication as described above. The ST/SC reaction was carried out by mixing 10 mL plant extract and 10 mL cell lysate overnight, at 4 °C, followed by centrifugation (3700× *g*, 20 min, 4 °C) and purification by passage through a 0.45 µm syringe filter (Carl Roth) followed by sucrose density centrifugation on a cushion of 2 mL 70% sucrose and 10 mL 25% sucrose [49]. Following ultracentrifugation (167,000× *g*, >3 h, 4 °C), the purified PVX-ST/SC-fusion protein particles were used for interaction analysis.

### 2.4. In Vitro Interaction Analysis—LOVTRAP and BphP1/QPAS1 Systems

Protein interaction in vitro was analyzed by discontinuous native PAGE using 4% stacking gels and 7.5% resolving gels, and NativeMark standards (Thermo Fisher Scientific). We mixed 5 µg of the purified proteins and incubated them for 30 min, at 4 °C, under appropriate illumination (Table 3). A list of the used LEDs can be found in Appendix A. We then added native PAGE loading buffer (62.5 mM Tris-HCl pH 6.8, 30% (*w*/*v*) glycerol, 0.05% (*w*/*v*) bromophenol blue), loaded the samples and separated them at 75 V for at least 2 h in Tris-glycine buffer pH 8.3 (25 mM Tris, 2 M glycine). An LED array was placed in front of the gel chamber if necessary. The gels were stained with Coomassie Brilliant Blue as described above.

### 2.5. Optical Switching of Dronpa145N

The purified Dronpa145N protein was diluted in PBS to 25 µM. One mixture was stored in the dark, at 4 °C. Two other samples were exposed to cyan light (505 nm) for 2 h, at 4 °C. One was subsequently irradiated with violet light (405 nm) for 30 s, and the other was stored in the dark. All samples were then incubated for 30 min, at room temperature, in the dark and mixed with native PAGE loading buffer before the separation of 2-µg samples by native PAGE as described above.

### 2.6. Protein Interactions on the VNP Surface—LOVTRAP and BphP1/QPAS1 Systems

We mixed 15 µg of the PVX-ST/SC-fusion protein particles with at least a three-fold molar excess of the protein partner for 30 min, at 4 °C, while irradiating with the appropriate wavelength of light to promote complex formation or dissociation. Grids for transmission electron microscopy (TEM) were treated with plasma to ensure a uniformly hydrophilic surface [50,51] before illumination as above. The particles were coupled to the grids for 2 h, at room temperature. Free binding sites were blocked with 0.5% (*w*/*v*) bovine serum albumin (BSA) in PBS for 30–60 min. The grids were then incubated overnight with the polyclonal antibody against PVX as described above or a monoclonal antibody against mCherry (GeneTex, Irvine, CA, USA, diluted 1:50 in PBS) and subsequently with the gold-labeled secondary antibody for 2 h. The grids were contrasted with 1% (*w*/*v*) uranyl acetate or 1% (*w*/*v*) phosphotungstic acid. Between each step, the grids were washed with Milli-Q water. Samples were analyzed at the University Hospital RWTH Aachen, Institute of Pathology, Electron Microscopy Facility (Aachen, Germany).

### 2.7. Protein Interactions on the VNP Surface—Dronpa145N System

We mixed 20 µg PVX-ST/His6-SC-G4S-Dronpa145N and 60 µg His6-mCh-G4S-Dronpa145N for 2 h, at 4 °C, under cyan light (505 nm). The samples were then rinsed by centrifugation (3700× *g*, 30 min, 4 °C) with PBS (SC-fusion) or 50 mM Tris-HCl pH 8.5 + 10% (*v*/*v*) glycerol (mCherry-fusion) using Vivaspin 6 columns with a MWCO of 1 MDa (Sartorius-Stedim) in the dark before incubation again for 30 min under cyan light (505 nm). The fusion protein partners were mixed and irradiated for 30 s under weak violet light (405 nm). All samples were then incubated for 30 min in the dark, at room temperature. The grids were treated with plasma, coupled to the protein complexes, blocked and probed with antibodies using the procedures described above but under red light (630 nm), followed by incubation in the dark.

## 3. Results

### 3.1. LOVTRAP

The N-terminus of the LOV2 protein was fused to the SC domain to facilitate binding to the surface of PVX particles displaying the ST peptide. The formation of an isopeptide bond between GST-SC-LOV2 and PVX-ST was confirmed by SDS-PAGE and Western blot (Figure 2A). In addition to the GST-SC-LOV2 band (57.1 kDa) detected with antibodies that recognize GST, and the PVX-ST band (27.4 kDa) detected with antibodies that recognize the PVX coat protein, we detected an additional band representing the protein complex (~85 kDa) only in the coupling reaction lane, which could be detected with antibodies against GST and PVX. The coupling efficiency was 40.2 ± 1.5% (n = 3). When mixing the cell lysate and plant extract, the analysis of purified PVX-ST/GST-SC-LOV2 complexes revealed a similar coupling efficiency of 43.8 ± 8.3% (n = 2). This confirmed the efficient binding of GST-SC-LOV2 to the PVX-ST coat protein in vitro.

We then investigated the interaction between LOV2 and Zdk1 in vitro by native PAGE, comparing the individual proteins alone to the coupling reaction under conditions that promoted complex formation (darkness) or dissociation (blue light). Native PAGE in the dark revealed a strong signal representing the protein complex in the coupling reaction lane, whereas native PAGE under blue light revealed signals for the individual proteins but not the complex (Figure 3A). The LOV2 protein band in the coupling reaction lane was also slightly more intense and the molecular weight was higher than the same band in the GST-SC-LOV2 lane. In addition to these bands, we observed fainter bands toward the top of the gel representing dimeric forms of the LOV2 protein. In the absence of Zdk1, the size of this band was ~450 kDa. In the coupling reaction lane, and only under dark conditions, the dimer band increased in size to ~480 kDa, which provided additional evidence confirming the protein interaction in the dark.

The loading of LOV2-modified PVX particles with His_6_-Zdk1-mCherry was analyzed by gold labeling and TEM (Figure 3B,C). Electron micrographs of the reaction under dark conditions showed the clear decoration of the particles with gold, confirming the interaction between the PVX-ST/GST-SC-LOV2 complex and His_6_-Zdk1-mCherry on the particle surface (Figure 3B and Appendix A). As expected, gold decoration was not observed under blue light because this wavelength promotes the dissociation of the LOV2-Zdk1 complex (Figure 3C and Appendix A). As an additional control, the particles were analyzed in the dark and under blue light without adding the His_6_-Zdk1-mCherry protein partner. The absence of gold labeling confirmed that the anti-mCherry/anti-His_6_ antibodies did not bind the modified PVX particles lacking the interaction partner (Appendix A). The blue light also had no effect on wild-type PVX particles (Appendix A).

### 3.2. BphP1/QPAS1

The N-terminus of the QPAS1 protein was also fused to the SC domain to facilitate binding to the surface of PVX particles displaying the ST peptide, and the formation of the SC/ST bond was confirmed by SDS-PAGE and Western blot as described above for the LOVTRAP system (Figure 2A). We observed bands for the individual GST-SC-QPAS1 protein (57.4 kDa) and PVX-CP coat protein (27.4 kDa) as anticipated, and in the coupling lane, we observed an additional band representing the assembled complex (~85 kDa), which again was detected in Western blots with both primary antibodies (Figure 2A). The coupling efficiency was 43.1 ± 11.3% (n = 3). When mixing the cell lysate and plant extract, the analysis of the purified PVX-ST/GST-SC-QPAS1 complex revealed a similar coupling efficiency of 44.8 ± 11.9% (n = 2). Native PAGE was carried out under illumination conditions that promoted complex formation (NIR) or dissociation (red light or darkness). The comparison of lanes containing the individual proteins and the coupling reaction revealed a high-molecular-weight band (~800 kDa) in the coupling reaction lane under NIR light that was not present in the individual protein lanes and only as a very weak signal in the coupling reaction lane under red light or in darkness (Figure 4A).

TEM analysis confirmed the interaction between BphP1 and QPAS1 attached to the modified PVX particles (Figure 4B,C). The pure PVX-ST/GST-SC-QPAS1 particles were mixed with BphP1-mCherry-His_6_ and incubated under NIR light (780 nm) or red light (630 nm), or in the dark. The particles irradiated with NIR light were decorated with gold, confirming the protein interaction on the particle surface (Figure 4B and Appendix A), whereas no labeling was observed under red light or in the dark (Figure 4C and Appendix A). Again, controls lacking the BphP1-mCherry-His_6_ protein also showed no evidence of gold labeling under any conditions, confirming that the anti-mCherry/anti-His_6_ antibodies did not bind to the modified PVX particles without the interaction partner (Appendix A). NIR and red light had no effect on wild-type PVX particles (Appendix A).

### 3.3. Dronpa145N

Finally, the reaction between His_6_-SC-G4S-Dronpa145N and PVX-ST particles was confirmed by SDS-PAGE and western blot analysis (Figure 2B). In addition to the His_6_-SC-G4S-Dronpa145N band (42.4 kDa) detected with antibodies that recognize the His_6_ tag, and the PVX-ST band (27.4 kDa) detected with antibodies that recognize the PVX coat protein, an additional band representing the protein complex (~70 kDa) was observed only in the coupling reaction lane, and was detected by antibodies against the His_6_ tag and the PVX coat protein. The coupling efficiency was 33.2 ± 6.2% (n = 3). When mixing the cell lysate and plant extract, the analysis of purified PVX-ST/His_6_-SC-G4S-Dronpa145N complexes revealed an increase in the coupling efficiency to 63.7 ± 14.3% (n = 2).

The photo-switchable oligomerization of Dronpa145N was also analyzed by native PAGE [52]. Dronpa145N was predominantly tetrameric in the dark. Irradiation with cyan light caused a switch to the monomeric state, but weak violet light restored the tetrameric state (Figure 5A). Calculation with ImageJ showed an initial total amount of tetrameric protein of 85%, which decreased to 63% by irradiation with cyan light and increased to 76% after illumination with violet light. These results show that the reversible light-dependent behavior of Dronpa145N is not affected by the presence of an N-terminal SC domain or ST peptide. We also confirmed the switch between the monomeric and tetrameric forms of His_6_-mCh-G4S-Dronpa145N (data not shown).

The ability of Dronpa145N to switch between oligomeric forms while attached to the PVX particle surface was confirmed by TEM (Figure 5B). Given the band intensity in the native gel (Figure 5A), we anticipated the presence of mainly tetrameric Dronpa145N on the particle surface after ST-SC coupling. The PVX-ST/His_6_-SC-G4S-Dronpa145N particles were therefore irradiated with cyan light to dissociate the tetramers before mixing with His_6_-mCh-G4S-Dronpa145N. We then irradiated the mixture with violet light to promote the formation of tetramers from the His_6_-SC-G4S-Dronpa145 and His_6_-mCh-G4S-Dronpa145N proteins. This was confirmed by using gold-conjugated anti-mCherry antibodies to detect His_6_-mCh-G4S-Dronpa145N displayed on the particle surface (Figure 5B). Only particles exposed sequentially to both light sources were abundantly decorated with gold. The micrograph shows several particles next to each other clearly labelled with gold (Figure 5B). When the PVX-ST/His_6_-SC-G4S-Dronpa145N particles were mixed with His_6_-mCh-G4S-Dronpa145N and exposed only to violet light without prior a cyan irradiation step, the gold labeling was very scattered (Appendix A). When PVX-ST/His_6_-SC-G4S-Dronpa145N and His_6_-mCh-G4S-Dronpa145N were irradiated with cyan light and mixed but not subsequently exposed to violet light, no gold labeling was observed (Appendix A). Mixing the components without irradiation also resulted in the absence of gold labeling (Appendix A). Samples lacking mCherry-Dronpa145N also showed no evidence of gold labeling, confirming that the anti-mCherry antibody did not bind to the modified PVX particles in the absence of His_6_-mCh-G4S-Dronpa145N (Appendix A). Neither violet nor cyan light affected the morphology of wild-type PVX particles (Appendix A).

## 4. Discussion

Light is ideal for the regulation of in vivo molecular processes such as drug release because it offers a high degree of spatial and temporal control, mainly due to the absence of intrinsic light sources in the body [1,53]. Alternative triggers such as pH and temperature are subject to natural variation between individuals and between different parts of the body. Thus far, most light-driven processes have relied on excitation with UV light, which is highly energetic and thus potentially harmful [54], despite its poor tissue penetration [55]. The development of systems based on visible light therefore offers a safer modality for controlled drug release.

The three optogenetic systems described herein allow the light-dependent formation and dissolution of protein complexes, which can be exploited to release drugs or imaging molecules from a larger macromolecular carrier. Plant viruses are suitable in this context because they provide a molecular scaffold for the external display of peptides and small proteins, as well as an internal cavity that can carry a molecular cargo. The N-terminus of the PVX coat protein is located on the outside of the viral capsid, so peptides and small proteins are easy to display on the virus surface by genetic fusion [56,57]. However, the displayed proteins or peptides must meet certain conditions to ensure the efficient self-assembly of VNPs: the isoelectric point must be in the range 5.2–9.2, and the size is limited to ≤8.5 kDa for direct coat protein fusions [58,59]. For larger proteins, the 2A sequence from foot-and-mouth disease virus (FMDV) can be inserted between the target gene and the coat protein gene, allowing the assembly of hybrid VNPs containing both wild-type coat proteins and fusions [60]. We considered both approaches for the creation of a photo-switchable release system using PVX particles, but neither was successful. The LOV2 protein cannot be fused directly to the N-terminus of the PVX coat protein because a free C-terminus is required to bind Zdk1 [8], and BphP1 (~80 kDa) is too large to support VNP assembly. However, even the proteins that were theoretically compatible with direct fusions did not support VNP assembly, and no hybrid particles formed when we used the FMDV 2A sequence or alternative 2A sequences (data not shown). We also considered chemical conjugation to exposed lysine or cysteine side chains as an alternative, but these methods are time consuming, the reactions cannot be directed precisely, and the chemicals involved can interfere with the native conformation of optogenetic proteins [38,61,62].

The remaining option was ST/SC technology, a plug-and-display system based on the formation of isopeptide bonds between the SC domain and the 13-amino-acid ST peptide, which are genetically fused to different proteins. A recombinant PVX coat protein including the ST peptide can therefore be used as a handle to attach any protein fused to the SC domain, resulting in the display of the SC-fusion protein on PVX particles [48]. We confirmed the formation of the isopeptide bond between PVX-ST and the SC-fusion proteins representing all three optogenetic systems. SDS-PAGE and Western blot analysis showed a mobility shift to ~85 kDa for the GST-SC-LOV2 and GST-SC-QPAS1 proteins, and to ~70 kDa for His_6_-SC-G4S-Dronpa145N (Figure 2). The fact that these run at higher molecular weight than expected can be explained by the nature of the PVX CP, which is slightly glycosylated by the plants [63,64]. A number of additional low-molecular-weight bands were also detected, probably representing degradation products. The coupling efficiency was 33–43% using the purified components and up to 63% when mixing cell lysates with plant extracts, which is comparable to the previously reported coupling of Cel12A-SC-His_6_ to PVX-ST [48] and within the 22–80% efficiency range typical for the ST/SC system [65]. PVX particles offer a large number of potential attachment sites due to the presence of 1270 coat protein subunits and the large overall surface area [48]. However, Thrane et al. (2016) reported a negative correlation between the size and density of antigens presented on the surface of spherical particles. If this also holds true for flexible particles, a coupling efficiency of up to 60% for a ~60 kDa protein indicates good efficiency. In comparison, the efficiency of chemical conjugation to viral coat proteins varies widely. For example, the fluorescent dyes 0488 and A647 were conjugated to lysine residues on PVX particles with coupling efficiencies of 70% and 90%, respectively [66], whereas the conjugation of 0488 to PVX cysteine residues resulted in a coupling efficiency of just 15% [67].

By coupling one protein partner to the PVX surface and fusing the other to mCherry, we were able to provide initial hypothesis of the in vitro protein interactions by native PAGE. For the LOVTRAP system, we observed a mobility shift in the protein signal for the complex when the experiment was run in the dark but not when exposed to blue light (Figure 3A). Zdk1 binds to LOV2 in the dark and is released under blue light due to a conformational change in LOV2 s [8,10]. Our native PAGE experiment therefore hypothesizes the system was functioning as anticipated. Interestingly, the individual LOV2 protein band also shifted to a higher molecular weight in the coupling reaction lane under blue light, probably reflecting differences in charge and conformation in the mixture compared to LOV2 on its own (Eubel et al., 2005). We also observed a higher-molecular-weight band representing a LOV2 dimer [68]. This also showed a mobility shift in the dark but not under blue light, giving further evidence of the formation of the anticipated LOV2-Zdk1 complex. The in vitro functionality of the LOVTRAP system was previously demonstrated using native PAGE by [47].

TEM and gold labeling confirmed the ST/SC reaction and particle loading. Overall, some viruses seem to be very blurry, but this can be explained by the addition of high amounts of glycerol to the samples, which is advantageous for the native structure of proteins [69] and is frequently stained by the uranyl acetate or phosphotungstic acid.

The PVX-ST/GST-SC-LOV2 particles were successfully loaded with His_6_-Zdk1-mCherry, as revealed by dense gold labeling following detection with anti-His_6_ and anti-mCherry antibodies (Figure 3B and Appendix A). The labeling was not visible under blue light, confirming that particle unloading is also possible (Figure 3C and Appendix A). Further controls showed that antibody binding does not occur in the absence of His_6_-Zdk1-mCherry (Appendix A) and that irradiation with blue light does not affect particle morphology (Appendix A). Gold particles were also present in the background, reflecting the adsorption of free His_6_-Zdk1-mCherry to the grids. Our results confirm that the LOVTRAP system retains its function on the PVX surface, allowing the formation of complexes in the dark state and their dissolution under blue light [8]. Although previous studies have used the LOVTRAP system to control cell signaling or protein activity in vivo [70,71], this is the first study to demonstrate the functionality of the system on the surface of nanoparticles.

For the BphP1/QPAS1 system, the mobility shift representing the formation of a complex was observed following exposure to far-red/NIR light but was nearly abolished under red light or in the dark (Figure 4A). Only a weak band at the level of the coupling product can be seen, accordingly it can be concluded that minimal amounts of coupling product can also be formed under red light. The Pfr from of BphP1 can interact with QPAS1 under NIR light [16], but QPAS1 is released when the complex is exposed to red light or darkness [14]. Our native PAGE experiments were therefore in general consistent with the formation and dissolution of the complex as anticipated. The BphP1 signal was very weak in the absence of its partner and cannot be shown in the complex, probably reflecting the conformational change of the protein under red light and in the dark, thus preventing the visualization of a clear signal. Thus, using native PAGE, initial hypotheses can be formulated regarding the function of BphP1/QPAS1 system. Signals can be found for both complex formation and dissociation, which can be interpreted accordingly.

TEM and gold labeling confirmed that the PVX-ST/GST-SC-QPAS1 particles could be loaded with BphP1-mCherry-His_6_ following irradiation with far-red light, as indicated by the abundant immunogold decoration following detection with gold-conjugated antibodies specific for His_6_ (Figure 4B) and mCherry (Appendix A). Conversely, no immunogold decoration was observed on particles exposed to red light or incubated in the dark (Figure 4C and Appendix A). As stated above for the LOVTRAP system, additional gold particles were observed in the background and can be attributed to the molar excess of BphP1-mCherry-His_6_. Further controls showed that antibody binding does not occur in the absence of BphP1-mCherry-His_6_ (Appendix A) and that neither far-red nor red light affects the particle morphology (Appendix A). The BphP1/QPAS1 system has previously been used to control gene expression and protein activity in vivo [5,16,72] but again this is the first study to demonstrate the functionality of BphP1/QPAS1 on the surface of nanoparticles.

Dronpa145N exists as a tetramer under violet light but dissociates into monomers under cyan light [73]. Our native PAGE experiments and densitometric analysis showed that cyan light diminished the tetramer band whereas violet light restored it, confirming that the system functions in vitro as anticipated (Figure 5A). However, the switch back to the tetrameric form was incomplete because the signal representing the reconstituted tetramer was weaker than that of the naïve complex. This may reflect steric hindrance caused by the SC or mCherry domains given that the switch between the monomeric and tetrameric forms of His_6_-Dronpa145N was previously shown to be almost complete [52].

TEM and immunogold labeling confirmed that PVX-ST/His_6_-SC-G4S-Dronpa145N particles could be successfully loaded with His_6_-mCh-G4S-Dronpa145N, but only following irradiation with cyan and violet light in succession (Figure 5B). This is because the naïve particles are already in the tetrameric state, which would prevent the formation of complexes with the free His_6_-mCh-G4S-Dronpa145N protein even when exposed to violet light (Appendix A). Accordingly, dissociation under cyan light is required before the mixing of PVX-ST/His_6_-SC-G4S-Dronpa145N and His_6_-mCh-G4S-Dronpa145N and exposure to violet light to promote hetero-tetramer assembly. Under these conditions, we detected particles displaying His_6_-mCh-G4S-Dronpa145N by immunogold labeling with antibodies specific for mCherry (Figure 5B). As discussed above for the other two systems, additional gold particles were observed in the background and can be attributed to the molar excess of His_6_-mCh-G4S-Dronpa145N. Even weaker background labeling was visible in the samples exposed to violet light without a prior dissociation cycle under cyan light, and this reflects the presence of a small number of monomers among the large excess of tetramers in the original preparation, enabling a small number of new tetramers to form under violet light (Appendix A). In contrast, exposure to cyan light without subsequent violet light irradiation resulted in no immunogold labeling at all, which we expected because the formation of new tetramers is dependent on the latter stimulus (Appendix A). Accordingly, we also observed no labeling when the sample was kept entirely in the dark (Appendix A). The morphology of the particles was unaffected by either light source (Appendix A). Dronpa145N has previously been used to crosslink light-controllable hydrogels [6] but this is the first use of the Dronpa145N system on the surface of nanoparticles.

The three optogenetic systems offer different advantages and disadvantages. The BphP1/QPAS1 system is particularly suitable for biomedical applications because infrared light can penetrate deep into tissue, and unlike UV light it is harmless to humans [53]. The reversal of the interaction in the dark also simplifies the release of cargo. However, the large size of BphP1 (~80 kDa) prevents direct fusion to the PVX coat protein due to steric hindrance [59] and it will therefore be challenging to increase the density of the protein on the VNP surface. The LOVTRAP system is advantageous in this context because both components are small (LOV2 = ~20 kDa and Zdk1 = ~6 kDa) making them more compatible with plant viruses as long as the C-terminus of LOV2 remains free [8]. However, blue light achieves significantly lower tissue penetration than infrared light [53]. Dronpa145N is a single-component system and is therefore simpler than the others, allowing the packaging and release of cargo by switching between tetrameric and monomeric forms. In the material sciences, this switch has been used to change the properties of hydrogels [6], and recent studies have also explored the benefits of plant viruses as components of hydrogels either for the presentation of biological signals that improve cell adhesion [74,75,76] or even as the structural basis of the hydrogel matrix itself [77]. By establishing light-switchable systems on the surface of plant viruses, the benefits of optogenetic systems and plant viruses can be combined in highly innovative therapeutic approaches.

## 5. Conclusions

Protein-based nanoparticles have many advantages over their synthetic counterparts in terms of biocompatibility, biodegradability and the precision made possible by genetically encoded domains with specific functions. This allows the development of more efficacious therapeutics with fewer and less-severe off-target effects, including carriers for the targeted delivery of drugs. The controlled release of drugs in terms of dose, timing and location is still a major challenge [78], but nanoparticles have the ability to deliver drugs in a controlled and noninvasive manner [53]. We have provided the first proof-of-concept for the applicability of photo-switchable proteins (in this case LOV2/Zdk1, BphP1/QPAS1 and Dronpa145N) on plant virus particles and have demonstrated its general in vitro functionality. When looking at the individual systems with their respective advantages and disadvantages, the BphP1/QPAS1 system is the most promising. Although the small molecular weight of the Zdk1 and LOV2 proteins reduces the possibility of steric hindrance and provides many application options, based on the obtained results, no disadvantage in terms of steric effects can be found for the BphP1/QPAS1 system. Similar efficiencies for coupling to the PVX-CP could be obtained for both systems, and also, based on the gold labeling, the LOV2 system did not perform better, although the BphP1 has about four-fold the molecular weight of the LOV2 protein. Complex formation between BphP1 and QPAS1 occurs using far-red light, which can penetrate deep into the tissue and is harmless compared to UV light. The reversal of the interaction in the dark also simplifies a later release of a cargo. Both the LOVTRAP and Dronpa145N systems are excited in the lower wavelength range, limiting application to upper skin layers. Therefore, the BphP1/QPAS1 system seems to be the most suitable, especially when it comes to biomedical applications such as drug release.

These initial findings provide the basis for the development of photo-switchable drug release systems based on optogenetic proteins on the surface of VNPs. The possibility to control these nanoparticles with light, independent of physiological parameters, lays the groundwork for a new range of controllable nanoparticle-based drug release systems.

## Figures and Tables

**Figure 1 biotech-11-00049-f001:**
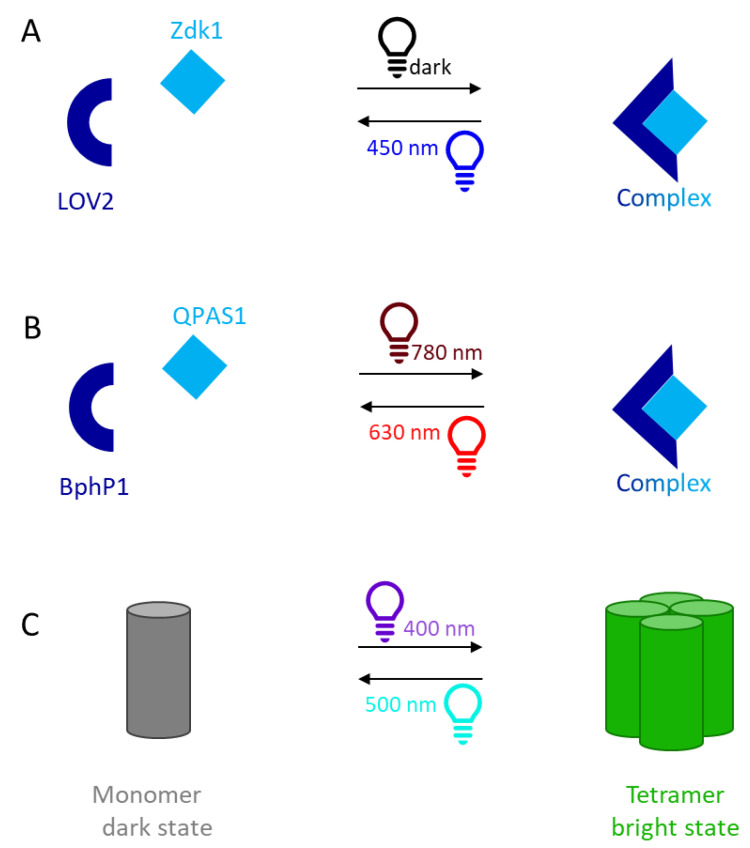
Schematic representation of three photo-switchable systems. (**A**) The Zdk1 protein binds to dark-state LOV2 and is released by irradiation with blue light. The binding and dissociation of the Zdk1 is dependent on the conformation of LOV2. LOV2 = light–oxygen–voltage 2, Zdk1 = Zdark1. (**B**) The QPAS1 protein binds to BphP1 when exposed to near-infrared light and dissociates when exposed to red light (or darkness, not shown). BphP1 = bacterial phytochrome photoreceptor P1 from *Rhodopseudomonas palustris*, QPAS1 = engineered partner for BphP1. (**C**) The single-component system Dronpa145N undergoes light-dependent oligomerization and dissociation. Excitation with cyan light favors the monomeric protein form, whereas violet light causes the protein to assemble into tetramers. This change in oligomerization state is accompanied by a change in protein fluorescence.

**Figure 2 biotech-11-00049-f002:**
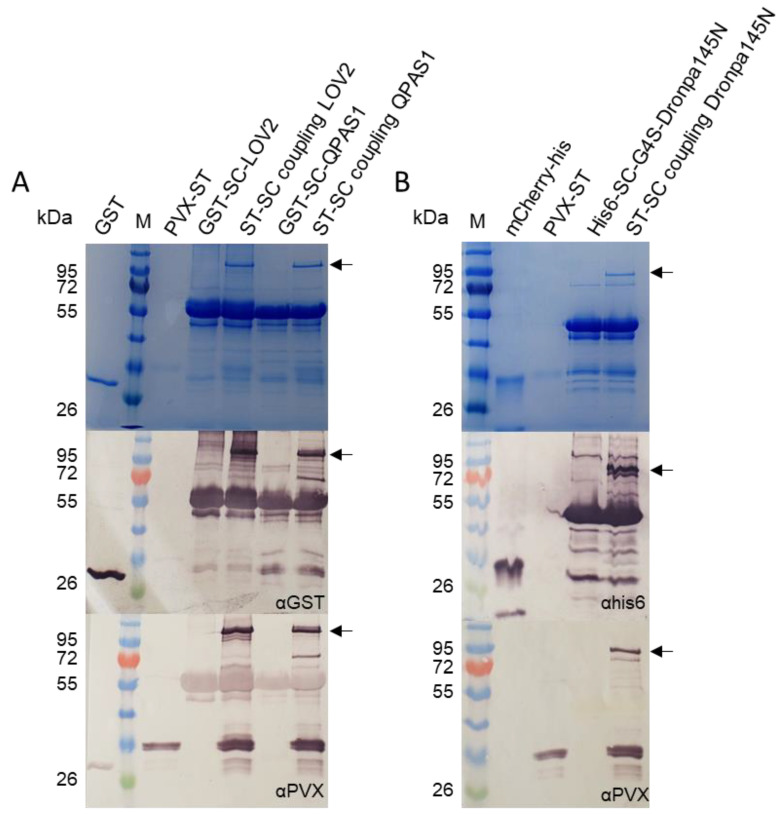
Coupling of purified optogenetic proteins to purified PVX-ST particles. (**A**) Conjugation of LOV2/QPAS1 to the surface of PVX particles using the ST-SC system confirmed by SDS-PAGE and Western blot. In addition to the bands for the individual GST-SC-LOV2 (57.1 kDa), GST-SC-QPAS1 (57.4 kDa) and ST-CP (27.4 kDa) proteins, a coupling band of ~85 kDa is present in Western blots probed with anti-GST and anti-PVX antibodies. (**B**) Conjugation of His_6_-SC-Dronpa145N to the surface of PVX particles using the ST-SC system confirmed by SDS-PAGE and Western blot. The anti-His_6_ antibody detected the monomeric Dronpa145N protein (42.4 kDa) and a larger protein representing the coupling reaction (70 kDa) which was also detected by the anti-PVX antibody. M = P7719 Color Prestained Protein Standard. Purified GST (28 kDa), PVX-ST (27 kDa) and mCherry-His_6_ (27 kDa) were loaded as controls. We mixed 3 µg of PVX-ST with a two-fold molar excess of the SC-fusion protein before loading. The arrows indicate the assembled complexes: PVX-ST/GST-SC-LOV2 = 85.1 kDa, PVX-ST/GST-SC-QPAS1 = 85.4 kDa, PVX-ST/His_6_-SC-G4S-Dronpa145N = 69.2 kDa.

**Figure 3 biotech-11-00049-f003:**
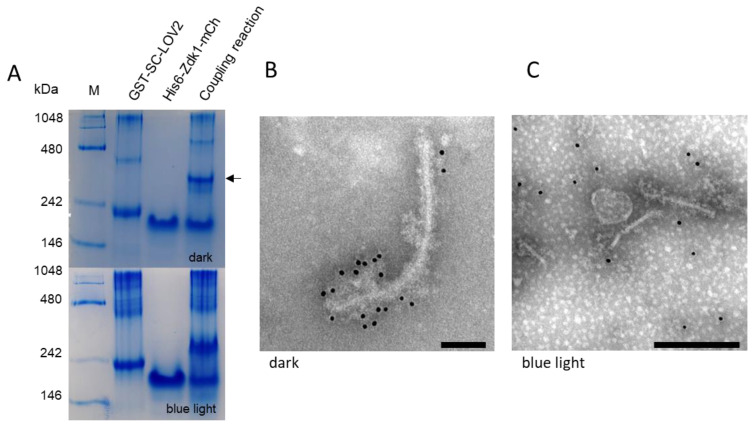
Analysis of the modified LOVTRAP system. (**A**) Interaction between the purified LOV2 and Zdk1 proteins analyzed by discontinuous native PAGE. In the dark, the bands representing the individual proteins GST-SC-LOV2 and His_6_-Zdk1-mCherry are joined by a higher-molecular-weight band (indicated by an arrow) representing the complex, which is not present under blue light. M = NativeMark standards. (**B**) Transmission electron micrograph showing the LOVTRAP system attached to PVX particles in the dark. Purified PVX-ST/GST-SC-LOV2 particles were mixed with His_6_-Zdk1-mCherry and incubated for 30 min in the dark. Particles were attached to the grids and His_6_-Zdk1-mCherry was detected with an anti-His_6_/GAM-12 nm gold conjugate. Immunogold decoration was observed only for the mixture of PVX-ST/GST-SC-LOV2 + His_6_-Zdk1-mCherry in the dark. (**C**) Transmission electron micrograph showing purified PVX-ST/GST-SC-LOV2 particles mixed with His_6_-Zdk1-mCherry. The mixture was incubated for 30 min in the dark. Particles were attached to the grids under blue light (455 nm) and His_6_-Zdk1-mCherry was detected with an anti-His_6_/GAM-12 nm gold conjugate. No immunogold decoration was observed on the particles. Scale bar = 100 nm (**B**), 250 nm (**C**). Controls are shown in Appendix A.

**Figure 4 biotech-11-00049-f004:**
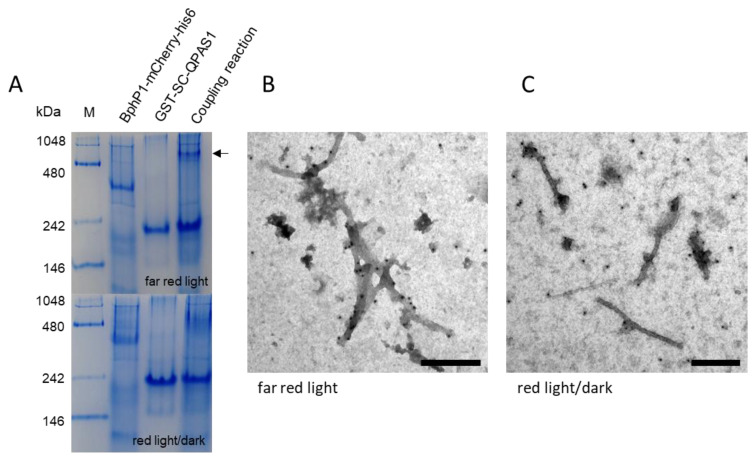
Analysis of the modified BphP1/QPAS1 system. (**A**) Interaction between the purified BphP1 and QPAS1 proteins analyzed by discontinuous native PAGE. Under far-red light, the bands representing the individual proteins BphP1-mCherry-His_6_ and GST-SC-QPAS1 are joined by a higher-molecular-weight band (indicated by an arrow) representing the complex, which is not present under red light. M = NativeMark standards. (**B**) Transmission electron micrograph showing the BphP1/QPAS1 system attached to PVX particles. Purified PVX-ST/GST-SC-QPAS1 particles were mixed with BphP1-mCherry-His_6_ and incubated for 30 min under far-red light (780 nm). Particles were attached to the grids and BphP1-mCherry-His_6_ was detected with an anti-His_6_/GAM-12 nm gold conjugate. Immunogold decoration was observed only for the mixture of PVX-ST/GST-SC-QPAS1 + BphP1-mCherry-His_6_ under far-red light (780 nm). (**C**) Transmission electron micrograph showing purified PVX-ST/GST-SC-QPAS1 particles mixed with BphP1-mCherry-His_6_. The mixture was incubated for 30 min in the dark. Particles were attached to the grids under red light or in the dark and BphP1-mCherry-His_6_ was detected with an anti-His_6_/GAM-12 nm gold conjugate. No immunogold decoration was observed for the particles. Scale bar = 250 nm. Controls are shown in Appendix A.

**Figure 5 biotech-11-00049-f005:**
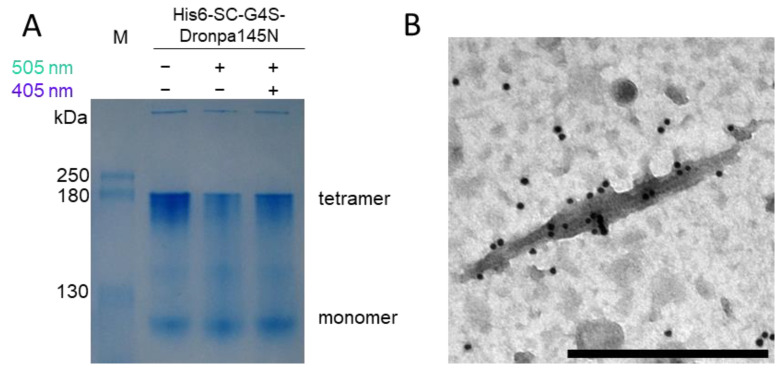
Analysis of the Dronpa145N system. (**A**) Analysis of the light-dependent association/dissociation behavior of Dronpa145N by discontinuous native PAGE. The monomeric state is favored under cyan light (505 nm) leading to the loss of signal intensity for the tetrameric form, but the latter is restored by exposure to weak violet light (405 nm) resulting in an increase in signal intensity. M = P7719 Color Prestained Protein Standards. (**B**) Transmission electron micrograph showing Dronpa145N attached to PVX particles. Purified PVX-ST/His_6_-SC-G4S-Dronpa145N particles were irradiated with cyan light (505 nm) before mixing with His_6_-mCh-G4S-Dronpa145N. The mixture was irradiated with violet light (405 nm) and particles were attached to grids. His_6_-mCh-G4S-Dronpa145N was detected with an anti-mCherry/GAM-12 nm gold conjugate. Immunogold decoration was observed for the mixture irradiated successively with cyan and violet light. Scale bar = 500 nm. Controls are shown in Appendix A.

**Table 1 biotech-11-00049-t001:** Expression conditions for the photo-switchable proteins.

Protein	Time [h]	Lighting Conditions	Temperature [°C]
His_6_-Zdk1-mCherry	16–18	Daylight	26
GST-SC-LOV2	16–18	Dark	26
GST-SC-QPAS1	16–18	Daylight	16
BphP1-mCherry-His_6_ ^§^	16–18	Dark	26
His_6_-SC-G4S-Dronpa145N	20–22	Dark	20
His_6_-mCherry-G4S-Dronpa145N	20–22	Dark	20

^§^ We added 30 µM biliverdin in 50 mM Tris-HCl pH 8.5 during induction.

**Table 2 biotech-11-00049-t002:** Buffers for lysis, washing and elution during the purification of each protein component. The LOV2 and Zdk1 proteins were purified as previously described ([47]). For the Dronpa145N proteins, all cell disruption and purification steps were carried out in the dark or under red light (630 nm) as previously described (Zhou et al., 2012).

Lysis Buffer	Wash Buffer		Elution Buffer
GST-SC-LOV2 and His_6_-Zdk1-mCherry			
50 mM Tris-HCl pH 8.5 150 mM NaCl 0.5% (*v*/*v*) Triton X-100 1 mM PMSF ^§^	Low salt: 50 mM Tris-HCl pH 8.5 150 mM NaCl 0.1% (*v*/*v*) Triton X-100 5 mM 2-mercaptoethanol ^§^	High salt: 50 mM Tris-HCl pH 8.5 650 mM NaCl 0.1% (*v*/*v*) Triton X-100 5 mM 2-mercaptoethanol ^§^	IMAC: 50 mM Tris-HCl pH 8.5 150 mM NaCl 150 mM imidazole Glutathione Sepharose: 25 mM reduced glutathione in low-salt wash buffer
BphP1-mCherry-His_6_			
50 mM Tris-HCl pH 8.5 300 mM NaCl 10% (*v*/*v*) glycerol	50 mM Tris-HCl pH 8.5 300 mM NaCl 10 mM imidazole 10% (*v*/*v*) glycerol		50 mM Tris-HCl pH 8.5 300 mM NaCl 300 mM imidazole 10% (*v*/*v*) glycerol
GST-SC-QPAS1			
PBS	PBS		50 mM Tris-HCl pH 8.5 25 mM reduced glutathione
His_6_-SC-G4S-Dronpa145N and His_6_-mCherry-G4S-Dronp145N			
25 mM Tris-HCl pH 7.8 300 mM NaCl 10 mM imidazole 10% (*v*/*v*) glycerol	25 mM Tris-HCl pH 7.8 300 mM NaCl 30 mM imidazole 10% (*v*/*v*) glycerol		25 mM Tris-HCl pH 7.8 300 mM NaCl 300 mM imidazole 10% (*v*/*v*) glycerol

^§^ Added immediately before use.

**Table 3 biotech-11-00049-t003:** Illumination conditions used in the in vitro binding assays and native PAGE experiments.

Optogenetic System	Direction	Incubation	Native PAGE
LOVTRAP	Binding	Dark	Dark
	Separation	Dark	455 nm (blue)
BphP1/QPAS1	Binding	780 nm (NIR)	780 nm (NIR)
	Separation	630 nm (red)/dark	630 nm (red)/dark

## Data Availability

Not applicable.

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
