# Peer review of "Plug-and-Display Photo-Switchable Systems on Plant Virus Nanoparticles"

_biotech, 2022, doi:10.3390/biotech11040049_

Round 1

Reviewer 1 Report

This manuscript reports the display of photo-switchable proteins on the surface of PVX. To achieve this, the authors use their previously developed Spy catcher/Spy tag (SC/ST) system to attached SC-labelled photo-switchable proteins to the surface of PVX particles in which the coat protein has been modified to display ST (PVX-ST).  The ability of the photo-switchable proteins to interact and release their binding partner under different light regimes.

While I found the experiments concerning the attachment of the photo-switchable proteins to the surface of PVX, as exemplified by Fig. 2, I was much less convinced by the experiments designed to show binding and release of the interacting partners. I have the following specific comments:

Figure 3A: One would expect treatment with blue light to give a banding pattern equivalent to simply mixing the two components. Indeed, this is precisely what is seen in Fig 1a of reference [47] cited in the current MS. However, this is not the case in Fig 3A, with an intense band appearing after blue treatment that migrates more slowly than the 242 kDa marker. The authors ascribe this to a change in conformation or charge of the LOV2 protein (lines 466-468) but present no evidence that this is its origin. Could it not, for example be caused by dimerisation of the Zdk1 protein under blue light conditions? The matter could be resolved by western blot analysis with appropriate antibodies as used in Fig. 2 and this should be done.

Figure 4A: This is not very convincing. There appears to be a somewhat smeared band at around 480kDa in the red light/dark treated sample as well as in the far red light samples. Indeed, the two gels look remarkably similar to me.

Figure 4B, C: The micrographs are not of publishable quality. There seems to be a lot of excess stain obscuring the viral particles and it is difficult to be sure what is going on with the gold binding.

Figure 5A: This seems to rely purely on visual inspection to determine the conversion of monomer to tetramer. I do not actually  see much difference in the ratio between the lanes. Some quantification, such as scanning the gels is definitely required to support the claim that ratio is different under different light conditions.

Figure 5B: It is difficult to make out what this represents - a single PV particle? A better image is required.

I also have the more general point that I found some of detail in the Materials & Methods section excessive and I would suggest this could be condensed by the use of references. I also found the the number of referenced cited (75) on the excessive size.

Author Response

General Answer: First of all, we thank the reviewers for critically reading and commenting on our manuscript. We need to point out that the presented data are first initial proof-of concepts to the system. We would have loved to test and develop the system further. However, due to structural changes in the institute (lab renovations, long term illnesses and personal change) we could not continue to work on the project. The corresponding author is no longer working at the university and the first author finished her thesis. The complete lab infrastructure is removed and not accessible for any more experiments. Nevertheless, we are convinced that the initial findings are worth publishing to allow other working groups to pick up on the idea and use our findings. We hope, the reviewers and editors see this as well and will accept the manuscript. We adapted the wording and enlarged the discussion. Therefore, we tried to be very precise in the method section, so other researchers can use and retest our findings. Additionally, we tried to discuss the available literature on every aspect available.

Reviewer: Figure 3A: One would expect treatment with blue light to give a banding pattern equivalent to simply mixing the two components. Indeed, this is precisely what is seen in Fig 1a of reference [47] cited in the current MS. However, this is not the case in Fig 3A, with an intense band appearing after blue treatment that migrates more slowly than the 242 kDa marker. The authors ascribe this to a change in conformation or charge of the LOV2 protein (lines 466-468) but present no evidence that this is its origin. Could it not, for example be caused by dimerisation of the Zdk1 protein under blue light conditions? The matter could be resolved by western blot analysis with appropriate antibodies as used in Fig. 2 and this should be done.

Answer: We thank the reviewer for the fruitful discussion. We in fact have tested to perform western blots from the native gels as suggest to prove our theory. However, the western blots with native gels as basis always led to a smear all over the lanes, with no distinct bands visual at all (also in every control we performed with the single proteins alone). The Zdk1 is shown do not undergo any conformational change upon blue light (or other light) stimulation, but LOV2 is described to do so and thus allows the binding of Zdk1. If the Zdk1 would dimerize this should be clearly visible in the controls.

We could show, that the band for the complex should not be present any more, but there should be a signal for the single LOV2 protein. We always observed that in blue native PAGE gels. Our discussion includes a possible shift for the LOV2 signal when mixed with Zdk1, although no complex is formed. This could be explained by the fact that the individual proteins run very close to each other. There is hardly any difference in the height of both bands. Mixing the proteins correspondingly shifts the signal for the LOV2 protein upwards. In addition, in native PAGE the molecular mass is not important, but above all the conformation and charge (Eubel et al., 2005). This can change when the two proteins are mixed, resulting in different separation behavior. Furthermore, a band for Zdk1 in the protein complex can be detected in the gel run in the dark. This might show that not all protein has bound to LOV2 and is therefore still freely available. In addition, the dimerization band of LOV2 gives further evidence for complex formation, since it is also shifted for the coupling reaction. This shift cannot be observed in the gel under blue light, both dimerization signals of LOV2 have the same molecular weight.

Reviewer: Figure 4A: This is not very convincing. There appears to be a somewhat smeared band at around 480kDa in the red light/dark treated sample as well as in the far red light samples. Indeed, the two gels look remarkably similar to me.

Answer: Thank you for the point. The native PAGE was used to get a general first impression regarding the function of the different systems, but the clearer proof of function was by TEM. It is true that the differences in native PAGE are only minimal. However, especially the small changes between gels that are important. The possible complex band at around 480 kDa is much more visible under far red light than under red light. Only a smear can be seen there, possibly indicating that small amounts of complex are also formed under red light, but do not result in a very clear band. We have tried to adjust this in our discussion. Also, we want to point out that this system was only used in vivo so far.

Reviewer: Figure 4B, C: The micrographs are not of publishable quality. There seems to be a lot of excess stain obscuring the viral particles and it is difficult to be sure what is going on with the gold binding.

Answer: We are sorry, that we could not create better images. We had large amounts of glycerol in the samples to stabilize the native proteins. The glycerol is always stained by the used TEM dyes. These blurry particles are often seen with plant viruses that are stabilized by additives. We added this to the discussion.

Reviewer: Figure 5A: This seems to rely purely on visual inspection to determine the conversion of monomer to tetramer. I do not actually  see much difference in the ratio between the lanes. Some quantification, such as scanning the gels is definitely required to support the claim that ratio is different under different light conditions.

Answer: We thank the reviewer for pointing this out. We included the calculations with ImageJ of the densitometric analysis of the lanes. The calculations show a change from ~86% tetramer to ~63 % tetramer and back to 77 %.

Reviewer: Figure 5B: It is difficult to make out what this represents - a single PV particle? A better image is required.

Answer: The picture shows several particles next to each other. Plant virus particles often tend to be found near each other in TEM images. Additional to the glycerol in the particles we needed to carefully select images, where the inner channel of the particles can be seen (like in the image provided). This picture was the clearest picture available. In comparison to plant virus pictures in general this picture shows a rather good impression. Higher quality pictures will hardly be obtained even after extensive improvements (that we can no longer test). We hope the reviewer accepts the explanation and expertise on plant viruses and their TEM images from our side.

Reviewer: I also have the more general point that I found some of detail in the Materials & Methods section excessive and I would suggest this could be condensed by the use of references. I also found the the number of referenced cited (75) on the excessive size.

Answer: We hope the explanation on top resolves this issue for the reviewer. Otherwise, we need some more time carefully check which references we find okay to remove. Additionally, we found it hard to refer to other publications regarding the methods because we had a very hard time figuring the optimal settings for the light switchable systems due to the only very briefly described methods elsewhere.

Reviewer 2 Report

This manuscript by Kauth et al. reports three plug-and-display photo-switchable systems on plant virus nanoparticles. These systems can undergo assembly and dissociation upon light stimulus, which might be applied in drug delivery. This is an interesting study, I recommend a minor revision. I have only two minor issues as follows:

1. Authors claim that the photo-swithable behavior of these systems is reversible, please kindly provide some evidences.

2. How about the binding and dissociation kinetics of these systems?

Author Response

General Answer: First of all, we thank the reviewers for critically reading and commenting on our manuscript. We need to point out that the presented data are first initial proof-of concepts to the system. We would have loved to test and develop the system further. However, due to structural changes in the institute (lab renovations, long term illnesses and personal change) we could not continue to work on the project. The corresponding author is no longer working at the university and the first author finished her thesis. The complete lab infrastructure is removed and not accessible for any more experiments. Nevertheless, we are convinced that the initial findings are worth publishing to allow other working groups to pick up on the idea and use our findings. We hope, the reviewers and editors see this as well and will accept the manuscript. We adapted the wording and enlarged the discussion. Therefore, we tried to be very precise in the method section, so other researchers can use and retest our findings. Additionally, we tried to discuss the available literature on every aspect available.

Reviewer: 1. Authors claim that the photo-swithable behavior of these systems is reversible, please kindly provide some evidences.

Answer: We were able to show the inversibility of the Dronpa system and added a densitometic analysis of the switch from tetramer to monomer and back. We included the calculations with ImageJ of the densitometric analysis of the lanes. The calculations show a change from ~86% tetramer to ~63 % tetramer and back to 77 %. The native PAGEs are only a initial analysis of the functionality of the system and we verified hypothesis that this would work on plant virus particles via TEM images. Due to the fixation of the particles and their dye it was not possible for us to show the reversibility of the system with the methods available. For a more in-depth analysis dynamic light scattering and calculation of zeta potential of the particles would have been needed. Unfortunately, we cannot provide this data. Otherwise, we did not state that the herein presented system is reversible because we could not deliver the data but only described that the photoswitchable function of the proteins is described as reversible in the introduction.

Reviewer: 2. How about the binding and dissociation kinetics of these systems?

Answer: We thank the reviewer for this idea. Currently no data is available for in vitro binding and dissociation kinetics of these systems. The idea we had to analyse the system in more detail was to use a FRET protein pair so we could use the fluorescence intensities to study the binding and dissociation kinetics. However, each easily usable FRET protein pair is also affected by the light sources needed to control the binding of the photoswitchable system. If we would have had the opportunity to further work on this project we would have loved to try to determine these kinetics. At least, there are calculated excitation and reversion times for the binding and dissociation that can be found on the optobase platform that we cited. We hope the reviewer accepts this explanation and finds the presented first proof of concept as important for the research community as we.

Reviewer 3 Report

Work done by Kauth et al. is interesting and new. However some major/minor edits are required to publish it in this reputed Journal.

Point wise comments are listed below…

1.      Abstract should be revised for proper flow.

2.      Introduction is too verbose; it should be concise and more informative.

3.      Objective of this study is not clearly mentioned.

4.      Line 60-61, should be revised.

5.      Line 114, Escherichia coli should be italic.

6.      Line 192, reference should be in one bracket only.

7.      Line 220, all references should be followed the same pattern, either name or number.

8.      Line 317-319 and 333, why highlighted?

9.      Spacing problem occurs within the text, correct it.

10          Grammar should be improved.

11 English language is not very good, check it.

Author Response

Point by point response Reviewer 3

Answer to points 10-11. Also 1 to 3!

 10          Grammar should be improved.

11 English language is not very good, check it.

  1. Abstract should be revised for proper flow.
  2. Introduction is too verbose; it should be concise and more informative
  3. Objective of this study is not clearly mentioned.

Answer: Honestly, we were very surprised by these issues. We always welcome a critical discussion of scientific work and find peer-reviews very important. Additionally, we find a clear and good description of our work of utmost importance. Therefore, we hired the Twyman Research Management company (https://www.twymanrm.com/) for the correction of the manuscript and to have a non-involved scientist critically reading the manuscript. The company is a scientific writing expert company (Specialist consultants in scientific project development, management, and presentation). Thus, we are very surprised, that the English and the structure is seen as a problem. We were working since several years with Richard Twyman and his company and never had this issue. If the editors see this as a problem as well, I would like to check back with Richard Twyman and discuss this matter with him further. The introduction was already condensed several times with the experts, but we find it very hard to further shorten it because we believe that hardly any reader will be familiar with all subjects that are important for this study (nanoparticles, plant viruses and optogenetics). Thus, we find it important to introduce every part sufficiently. If the editors also find the introduction to long, we can try to further shorten it, but then we need more time. Equally the abstract and objective of the study was several times rewritten with the support by the Twyman Research Management to give the best overview and structure. We have discussed several possibilities with the expert team and were all convinced that the present structure was the optimal one for this complex topic. The current reviewer is the first reviewer with this issue and another restructuring of the manuscript is currently not possible for us in a short timeframe.

  1. Line 60-61, should be revised.

Answer: we have rewritten the sentences.

  1. Line 114, Escherichia coli should be italic. and 6. Line 192, reference should be in one bracket only.

Answer: thanks for finding these formatting mistakes. We have corrected the format.

  1. Line 220, all references should be followed the same pattern, either name or number.

Answer: Thank you for finding this mistake. We have corrected the reference.

  1. Line 317-319 and 333, why highlighted?

Answer: These highlights were corrections for other reviewers and are removed now.

  1. Spacing problem occurs within the text, correct it.

Answer: We checked the spacing and corrected it wherever we found problems.

Round 2

Reviewer 1 Report

I have looked at the resubmitted version of the manuscript and have considered the authors' responses to my various comments carefully. I have great sympathy with the issues that can be caused by some authors leaving the institution where the work was carried out and laboratory facilities becoming unavailable. Unfortunately this happens quite frequently, leaving work incomplete and not suitable for publication.

The main issue is whether research which the authors themselves describe as "first initial proof-of-concepts to the system" is suitable for publication in a peer-reviewed journal at this point. My opinion is, that despite the revisions made to the MS, it does not meet the required standard. I therefore reluctantly recommend rejection unless further work can be done. However, I suggest that authors consider depositing the MS in and archive, such as BioRx, so that the preliminary information is available to scientific community if other groups wish to pursue the initial work.

Author Response

We thank the reviewer for the understanding. We decided to go for a peer-reviewed publication because "although" this is only an initial proof-of-concept, it is a very well tested proof with active proteins (fluorescent proteins). These results are the work of a three year PhD thesis and we tried to critically discuss the results so we believe the work is worth a peer-reviewed publication.